# Interaction of Soybean (*Glycine max* (L.) *Merr.*) Class II ACBPs with MPK2 and SAPK2 Kinases: New Insights into the Regulatory Mechanisms of Plant ACBPs

**DOI:** 10.3390/plants13081146

**Published:** 2024-04-19

**Authors:** Atieh Moradi, Shiu-Cheung Lung, Mee-Len Chye

**Affiliations:** School of Biological Sciences, The University of Hong Kong, Pokfulam, Hong Kong, China; moradi@hku.hk

**Keywords:** plant acyl-CoA-binding proteins, phosphorylation, kinases, yeast two-hybrid, bimolecular fluorescence complementation

## Abstract

Plant acyl-CoA-binding proteins (ACBPs) function in plant development and stress responses, with some ACBPs interacting with protein partners. This study tested the interaction between two Class II GmACBPs (*Glycine max* ACBPs) and seven kinases, using yeast two-hybrid (Y2H) assays and bimolecular fluorescence complementation (BiFC). The results revealed that both GmACBP3.1 and GmACBP4.1 interact with two soybean kinases, a mitogen-activated protein kinase MPK2, and a serine/threonine-protein kinase SAPK2, highlighting the significance of the ankyrin-repeat (ANK) domain in facilitating protein–protein interactions. Moreover, an *in vitro* kinase assay and subsequent Phos-tag SDS-PAGE determined that GmMPK2 and GmSAPK2 possess the ability to phosphorylate Class II GmACBPs. Additionally, the kinase-specific phosphosites for Class II GmACBPs were predicted using databases. The HDOCK server was also utilized to predict the binding models of Class II GmACBPs with these two kinases, and the results indicated that the affected residues were located in the ANK region of Class II GmACBPs in both docking models, aligning with the findings of the Y2H and BiFC experiments. This is the first report describing the interaction between Class II GmACBPs and kinases, suggesting that Class II GmACBPs have potential as phospho-proteins that impact signaling pathways.

## 1. Introduction

Acyl-CoA-binding proteins (ACBPs) are important in transporting acyl-CoA esters, which are the CoA-activated form of free fatty acids [1]. Additionally, they are a crucial housekeeping protein family [2] and are involved in signaling pathways and stress responses in eukaryotic cells, including *Caenorhabditis elegans* [3], mammals [4], and plants [5]. Plant ACBPs play various roles in growth and development [6,7] and have been well-characterized by their participation in responses to adverse environmental conditions, e.g., cold [8,9,10], pathogen infection [11], drought [12], salinity [13,14], heavy-metals [15,16], and hypoxia [17]. Over the past two decades, the multi-membered ACBP family in Arabidopsis [18], rice [13], *Brassica napus* [19], oil palm [20], and more recently in soybean [21], have been identified and characterized. Based on their molecular mass, domain architecture, and phylogeny, plant ACBPs have been grouped into four classes [13], namely small ACBPs (Class I), ankyrin-repeat ACBPs (Class II), large ACBPs (Class III) and kelch-ACBPs (Class IV). However, the lack of knowledge on the signaling functions and the regulation of post-translational modifications of plant ACBPs prevents a better understanding of their roles during harsh environmental stresses.

Soybean, highly valued for its protein content and health benefits, is susceptible to low yields due to its unfavorable environmental conditions. Therefore, it is essential to acquire a deeper understanding of the function of GmACBPs in response to extreme environmental conditions and their role in controlling signaling pathways and interaction with other proteins, particularly kinases. Although the importance of GmACBPs has been recognized, limited research has been carried out to investigate their functions. In a particular study, the use of confocal laser scanning microscopy demonstrated that EGFP fusion proteins of Class II GmACBPs were found to be present in the same location as the endoplasmic reticulum (ER)-Tracker, specifically at the membrane of vesicles derived from the ER. Additionally, the study revealed that the ligand-binding status of Class II GmACBPs plays a crucial role in determining their interaction with protein partners (e.g., 9-LIPOXYGENASE) [14]. This interaction, in turn, regulates the production of oxylipin signals in the roots of soybean plants in response to salinity [14]. In a recent investigation that utilized isotopically dimethyl labeling-based quantitative proteomic analysis, several phosphosites were identified in Class II GmACBPs [22]. The presence of these phosphosites implies that phosphorylated GmACBPs potentially contribute to the adaptation of soybean plants to drought [22]. Furthermore, it has been discovered that protein phosphorylation can regulate the acyl-CoA-binding pocket of a TetR-like transcription factor in the archaeal model organism *Sulfolobus acidocaldarius* [23].

The lack of pertinent knowledge on the signaling functions of plant ACBPs has prompted this study on the interaction of soybean Class II ACBPs with several kinases. Potential candidates of SNF-related serine/threonine-protein kinases (SnRKs), mitogen-activated protein kinases (MPKs), calcium-dependent protein kinases (CDPKs) and other protein kinases were tested for their interactions with Class II GmACBPs using *in vivo* methods, including yeast-two hybrid (Y2H) screening and bimolecular fluorescence complementation (BiFC) analysis. Phos-tag SDS-PAGE was used to determine the ability of kinases to phosphorylate Class II GmACBPs. To comprehend the protein kinase-regulated signaling pathways, it is crucial to identify and characterize kinases, as well as their distinct phosphorylation sites. Although most or all protein kinases have been identified, the specific sites they phosphorylate are not well-known. Numerous computational methods are available for predicting the phosphorylation sites [22]. To improve the precision of predicting phosphorylation sites for Class II GmACBPs, three protein phosphorylation databases were used herein to compile a collection of Class II GmACBP phosphorylation sites. Also, the HDOCK server was used to automatically predict the binding models of Class II GmACBPs with kinases.

## 2. Results

### 2.1. Interaction between Class II GmACBPs and Soybean Kinases

The significant homology between GmACBP3.1 and GmACBP4.1 is the primary reason for the identical results obtained from the Y2H assay (Figure 1A). Out of the seven protein kinase candidates that were tested in this study, the Y2H analysis revealed a strong interaction between Class II GmACBPs and GmMPK2, as well as one GmSAPK2 (Figure 1B). An additional Y2H assay was conducted to determine the specific domains of Class II GmACBPs, which are crucial for their interaction with GmMPK2 and GmSAPK2. The results indicated that for both kinases, the only interaction was observed with the ankyrin-repeat (ANK) domain, but not with the acyl-CoA-binding (ACB) domain (Figure 1C). Therefore, the interaction between the Class II GmACBPs and these kinases strictly depends on the presence of the ANK domain.

Interactions detected in yeast cells were verified using BiFC. Five combinations of proteins were co-expressed in *Nicotiana benthamiana* leaves. The results showed that significant YFP signals were generated in *N. benthamiana* cells when both combinations GmMPK2:cYFP/GmACBP4.1:nYFP (native form) and GmSAPK2:cYFP/GmACBP4.1:nYFP (native form) were co-infiltrated (Figure 2). Additionally, the positive control presented a robust signal, but no BiFC signals were observed for the splice variant (Figure 2); hence, the ANK domain of Class II GmACBPs was shown to be crucial for kinase interaction. Therefore, these findings validate the interaction of GmACBP4.1 with GmMPK2 and GmSAPK2 in living plant cells. It was concluded that the interaction criteria for GmACBP3.1 and MPK2, as well as GmACBP3.1 and SAPK2, are similar, as the two members of Class II GmACBPs exhibit high homology (Figure 1A). These results are consistent with the outcomes of the Y2H assay, and the findings from Y2H and BiFC experiments demonstrated that only the ANK domain activity of Class II GmACBPs is essential for their interaction with kinases.

Proteins from co-transfected yeast cells were used for Western blot assays, and the results confirmed the expression of all target fusion proteins in the GAL4 system (Figure 3A,B). Furthermore, additional Western blot assays demonstrated the expression of nYFP and cYFP-fusion proteins in *N. benthamiana* leaves (Figure 3C,D). This implies that the absence of interaction with kinases is the reason why the splice variant fails to reconstitute YFP.

Given that Class II GmACBPs were shown to interact with two selected protein kinases by Y2H and BiFC assays, an *in vitro* kinase assay was conducted to investigate whether Class II GmACBPs are the true substrates of these kinases. The resulting kinase reactions were analyzed using Phos-tag acrylamide SDS-PAGE, which allows for the separation of phosphorylated and non-phosphorylated GmACBP4.1. Upon analysis, a shift in size was observed for GmACBP4.1 after treatment with both GmMPK2 and GmSAPK2, indicating specific substrate phosphorylation by these kinases (Figure 4). To ensure the specificity of the observed phosphorylation, control reactions were included in the *in vitro* kinase assay, which solely contained GmACBP4.1 or kinases. Overall, the findings from Y2H, BiFC, and the *in vitro* kinase assay, combined with the distinct size shift observed on the Phos-tag acrylamide SDS-PAGE, provide strong evidence that Class II GmACBPs are indeed substrates for GmMPK2 and GmSAPK2 protein kinases.

### 2.2. Prediction of Kinase-Specific Phosphorylation Sites in Class II GmACBPs and Docking-Based Binding Models of Class II GmACBPs and Two Kinases

To improve the accuracy of identifying kinase-specific phosphorylation sites in Class II GmACBPs, the result of three prediction protein phosphorylation databases, including EPSD 1.0, GPS 6.0, and NetPhos 3.1, were integrated to generate a comprehensive collection of phosphorylation sites for Class II GmACBPs. The Appendix A present the outcomes of predicting kinase-specific phosphorylation sites in Class II GmACBPs for each database.

Of the 38 phosphosites predicted (Figure 5), 23 were found in the NetPhos database, ten in the GPS database and five in the EPSD database. The NetPhos database predicted more phosphosites for Class II GmACBPs (Figure 5B). Interestingly, nine of these phosphosites were common among the predictions from all three databases. These overlapping phosphosites include T49, S56, S57, S66, S78, S120, T140, S193, and S256 (Figure 5B). Upon further investigation of the phosphopeptides, it was observed that the number of phosphorylation events on serine residues was nearly twice that of threonine residues, with twenty phosphosites for serine and ten for threonine (Figure 5C,D). Additionally, the distribution of these phosphosites on Class II GmACBPs showed twenty-four phosphosites on the ACB domain and four on the ANK domain (Figure 5D).

The databases used in this investigation predicted the kinase families associated with the phosphosite-containing substrates (Figure 6). The predictions resulted in the identification of seven kinase families, which are AGC (cAMP-dependent protein kinase), CK (casein kinase), CDK (cyclin-dependent kinase), MPK (mitogen-activated protein kinase), GSK (glycogen synthase kinase), PIKK (phosphatidylinositol-3 kinase-related kinase) and CLK (Cdc2-like kinase). Commonly, prediction databases utilize patterns in the phosphosite regions to identify specific kinases likely to phosphorylate them [24]. The CK and AGC kinase families had the highest number of motifs, with 18 and 15 docking site motifs, respectively, among the potential kinases identified (Figure 6). It is important to note that a single phosphosite could be associated with multiple kinase families. These findings indicate that Class II GmACBPs could act as phospho-proteins, influencing the signaling pathways in response to challenging environmental conditions.

The HDOCK server was used to automatically predict the interaction between GmACBP3.1 and two kinases for protein–protein docking, generating binding models. Among the top 100 models for both complexes, the first models with the lowest docking scores and highest confidence scores are shown in Figure 7. Specifically, the first model of the GmACBP3.1-GmMPK2 complex has a docking score of −220.98 and a confidence score of 0.8053, while the first model of the GmACBP3.1-GmSAPK2 complex has a docking score of −229.37 and a confidence score of 0.8302. The HDOCK server employed evaluation metrics to determine the likelihood of binding between two molecules, where a lower docking score indicated a higher possibility of binding. A confidence score above 0.7 signified a high probability of binding. Furthermore, the HDOCK server also provided two types of binding site information as constraints for docking, namely binding site residues on the receptor (kinases) and ligand (GmACBP3.1) and distance restraints between the binding site residues of the receptor and ligand (Figure 7). Based on this binding site information, it was observed that affected residues were only present in the ANK region of GmACBP3.1 in both docking models, suggesting the crucial role of the ANK domain in protein–protein interaction.

## 3. Discussion

Of all post-translational modifications (PTMs), phosphorylation is the most common regulatory mechanism to alter protein conformation, modify protein functions, and transmit signals within cells [25,26]. Protein kinases and phosphatases play crucial roles in governing the process of protein phosphorylation. Understanding how protein phosphorylation events control cellular responses, particularly during various stresses, is an intriguing subject that can aid in the better comprehension of signaling pathways. To address this, the first step is to identify specific kinases and their functions during stresses. The lack of understanding of the regulation and signaling functions of post-translational modifications in plant ACBPs is a challenge in comprehending their involvement in harsh environmental stresses. To understand how these proteins mediate plant stress responses, it is essential first to identify ACBP interactors, particularly kinases.

In addition to plants, different species contain various ACBP homologous proteins with varying numbers of ACBPs. For example, there are seven types of ACBD family proteins in humans. ACBD1, or ACBP, is the smallest protein and solely consists of the ACBP domain. ACBD5, on the other hand, has 525 amino acid residues and exclusively contains the ACBP domain. The remaining five ACBD family proteins contain the ACBP domain and other domains like ANK, GOLD, and ECH [27]. ACBP, also referred to as the diazepam binding inhibitor (DBI), earned its name in 1983, when it was identified as a brain peptide which exhibits a strong affinity for benzodiazepine gamma-aminobutyric acid (GABA) receptors [28]. For the first time, Faergeman et al. (2002) showed the crucial function of ACBP in human cells. The researchers employed small interference RNA to silence *ACBP* in HeLa, HepG2, and Chang cells. Introducing ACBP-specific siRNA inhibited the growth and detachment of cells from the growth surface and prevented thymidine and acetate incorporation [4].

Apart from acting as a protein which stores and transports different acyl-CoAs in the ER, mitochondria, plasma membrane, and Golgi body, DBI also has a critical role in preventing the premature degradation of acyl-CoAs. This dual function ensures the availability of an adequate lipid pool in different organelles and facilitates the modification of lipids in cell membranes [29,30]. Recently, phosphorylated ACBD was shown to regulate nutrient-dependent autophagy [31]. In this report, ACBD interacted with phosphatidylethanolamine found in the phagophore membrane, hindering the lipidation of LC3 proteins [31]. This inhibition prevents the initiation of autophagy when nutrient-rich conditions are present. However, during periods of severe nutrient deprivation, ACBD undergoes phosphorylation by AMPK at serine-21. This phosphorylation event leads to the loss of ACBD’s affinity for phosphatidylethanolamine, causing it to detach from the phagophore membrane. Consequently, this release enhances LC3 lipidation, ultimately triggering autophagy initiation [31].

It is known that ACBPs play crucial roles in lipid metabolism, stress responses, and hormone signaling pathways in plants. Given the importance of protein–protein interactions in signaling pathways, we hypothesized that the interaction between Class II GmACBPs and kinases could potentially play a role in modulating these pathways. By investigating the interaction between Class II GmACBPs and specific kinases, we aim to gain insights into the potential involvement of plant ACBPs in signaling cascades and the underlying mechanisms that contribute to plant growth and development. An understanding of these interactions could provide valuable information for future research on manipulating signaling pathways and improving plant traits for various agricultural applications.

This study used several bait and prey plasmids for Y2H analysis to investigate the interaction between Class II GmACBPs and seven kinases (Figure 1). It utilized BiFC to confirm the identified interacting partners (Figure 2). Class II GmACBPs were found to interact with two soybean kinases, namely GmMPK2 and GmSAPK2. Also, this study showed that these two kinases can phosphorylate Class II GmACBPs *in vitro* (Figure 4). Experiments in Y2H, BiFC, and docking modeling studies demonstrated that the interactions were facilitated by the ANK domain. These interactions provide essential clues for a more comprehensive understanding of the role of plant ACBPs in signaling pathways.

GmMPK2, utilized for protein–protein interaction, is a homolog of *Arabidopsis thaliana* MPK6 (Table 1). The mitogen-activated protein kinase cascades have a wide range of functions in transmitting signals and stress responses in plants [32,33], including salt stress [34]. The activity of MPK6 is enhanced through its physical binding with phosphatidic acid [34,35], which is known to interact with Class II GmACBPs to regulate the response to salinity [14]. Additionally, Zhou et al. [35] found that MPK6 has a role in controlling the activity of RELATED TO AP2.12 [36] and PHOSPHOLIPASE Dα1 [37], which were identified as protein partners of Class II AtACBPs during hypoxia signaling.

Herein, the interaction between Class II GmACBPs and GmSAPK2, a homolog of *Arabidopsis thaliana* SnRK2.8 (Table 1), was validated. SnRK2.8 belongs to a group of plant-specific serine/threonine kinases called SnRKs, which are homologous to the AMPK/SNF1 family [38] and play a significant role as stress-related protein kinases in plants [39]. SnRK2.8 is essential in regulating plant metabolism and growth by phosphorylating and controlling the activity of various enzymes, including 60S ribosomal protein, 14-3-3 proteins, glyoxalase I, and adenosine kinase [40]. Additionally, SnRK2.8 plays a crucial role in stress tolerance, including drought [39,41] and osmotic stress [42]. Interestingly, these SnRK2 family members can phosphorylate AREB1 (ABA-RESPONSIVE ELEMENT BINDING TRANSCRIPTION FACTOR1) [43]. AREB1 is a protein partner of Class II AtACBPs [44], and it has been reported that the association between AREB1 and Class II AtACBPs can facilitate ABA signaling, particularly during germination and vegetative growth [37]. Furihata et al. [45] examined the phosphorylation of a recombinant AREB1 polypeptide by Arabidopsis SnRK2 members, such as SnRK2.8 using an in-gel protein kinase assay, employing SnRK2-GFP fusion proteins that were overexpressed in cultured cells of Arabidopsis T87. The results showed that all these SnRK2-GFP proteins could phosphorylate the AREB1 polypeptides [45]. These findings indicated that MPK2 and SAPK2 can interact with proteins associated with Class II ACBPs, suggesting that these kinases might also interact with ACBPs directly, phosphorylate them, and regulate their activity, particularly in response to different stress conditions.

In a previous investigation [22] in phosphoproteomics analysis using dimethyl labeling on soybean under drought conditions, a total of 279 kinases, including GmMPK2 and GmSAPK2, were observed with PSM (Peptide Spectrum Match) counts of 124 and 2, respectively. Additionally, following the mapping of phosphosites with kinase proteins, it was discovered that two phosphosites, belonging to Class II GmABPs, act as docking sites for soybean MPK2 [22]. This suggests that MPK2 has the potential to interact with Class II GmABPs and phosphorylate them under drought conditions.

Moreover, this research highlights the significance of the ANK domain in facilitating protein–protein interactions, especially interactions with kinases. Lung et al. [14] established that the ANK domain of Class II GmACBPs is essential for their interaction with VLXB, which is a homolog of LIPOXYGENASE (LOX). The ANK domain is also crucial for interaction of Arabidopsis Class II AtACBP1 with STEROL C4-METHYL OXIDASE1 [46,47]. The ANK repeat is a common protein motif which exists extensively in nature and is essential for mediating protein–protein interactions [48]. The ANK motif is present in Class II ACBPs [13], and previous studies have confirmed its involvement in protein–protein interaction [15,49,50]. These Y2H and BiFC experiments herein produced results that align with previous findings. For instance, one study demonstrated that Arabidopsis ACBP2 interacts with LYSOPHOSPHOLIPASE2 [49]. This interaction is facilitated by the ANK domain of AtACBP2, which is consistent with observations made in Y2H and co-immunoprecipitation assays. Another investigation found that AtACBP2 interacts with the *A. thaliana* ethylene-responsive element-binding protein through its ANK domain [50]. However, this interaction was absent in the Y2H analysis when the ANK domain was removed. The ANK domain has been previously shown to be necessary for the interaction between plant ACBPs and other proteins but not including kinases. Therefore, this study is the first to emphasize the significance of this domain in facilitating the interaction between plant ACBPs and kinases.

Apart from these discoveries, this study also made predictions of kinase-specific phosphorylation sites for Class II GmACBPs using three databases: GPS 6.0, EPSD 1.0, and NetPhos 3.1 (Figure 5 and Figure 6). Using computational methods to predict protein phosphorylation sites is a potentially practical approach for reducing costs and is time-associated with experimental techniques. Predicting phosphorylation sites can provide valuable insights into the molecular mechanisms of phosphorylation events and assist in the functional characterizing proteins. The prediction results indicated that a total of 38 phosphosites (Figure 5B) and 7 families of kinases were identified (Figure 6), and AGC and CK kinases exhibit the most significant number of docking site motifs on Class II GmACBPs (Figure 6). The AGC [51,52] and CK [53,54] protein kinase families are essential regulators of various developmental processes and plant stress responses. A phosphoproteomics study, conducted on soybean plants under drought conditions [22], revealed that by mapping phosphosites with kinase proteins, multiple protein kinase families, including CK, CLK, CDK, and MPK, were capable of phosphorylating Class II GmABPs. Upon analysis, it was discovered that certain kinase families predicted from this study coincide with the findings of a previous phosphoproteomics study on soybeans [22]. The HDOCK server was also used to predict the binding models of the GmACBP3.1-GmMPK2 and GmACBP3.1-GmSAPK2 complexes. The results showed that the affected residues were specifically situated in the ANK region of GmACBP3.1 in both docking models, indicating the critical role of the ANK domain in promoting protein–protein interactions.

The evidence suggests that MPK2 and SAPK2 kinases are essential in the signaling pathways of various plants, especially in soybean. Earlier studies [35,43,45] have demonstrated their interaction with protein partners of plant ACBPs such as AREB1 and RELATED TO AP2.12, while this study illustrates their interaction with Class II GmACBPs. This implies that these kinases have the potential to affect the function of Class II GmACBPs. These interactions provide new insights into the regulatory mechanisms of plant ACBPs. It suggests that these interactions may affect soybean growth, development, and response to environmental stresses, including biotic and abiotic factors. Furthermore, an understanding of these interactions holds potential implications for soybean breeding, genetic engineering, and crop improvement efforts.

**Table 1 plants-13-01146-t001:** The list of kinases used in this study for protein–protein interactions.

Name in *Arabidopsis thaliana*	Accession Number (NCBI)	*Glycine max* Homolog ^a^	Accession Number (NCBI)	Refs.
Mitogen-activated protein kinase 3	NP_190150.1	Mitogen-activated protein kinase 13	XP_003538034.1	[35,55]
Mitogen-activated protein kinase 6	NP_181907.1	**Mitogen-activated protein kinase 2**	NP_001235426.1	[34,35,55]
Snf1-related protein kinases SnRK2.8	NP_001077839.1.1	**Serine/threonine-protein kinase SAPK2-1**	XP_003519175.1.1	[39,42]
Snf1-related protein kinases SnRK2.7	NP_195711.1	Serine/threonine-protein kinase SAPK2-2	XP_003531338.1	[39,42]
Snf1-related protein kinases SnRK2.2	NP_190619.1	Serine/threonine-protein kinase SRK2I	XP_003550077.3	[56]
Casein kinase II, Beta chain 1	NP_001190483.1	Putative casein kinase II subunit beta	NP_001344382.1	[53,57,58]
Calcium-dependent protein kinase 4)	NP_192695.1	Calcium-dependent protein kinase 4	XP_006589459.2	[59]

^a^ The two kinases, which interactions with Class II GmACBPs were confirmed in this study, were highlighted in bold.

## 4. Materials and Methods

### 4.1. Plasmids Construction for Yeast Two-Hybrid Assay

Based on the pertinent information from the literature [34,35,39,42,53,55,56,57,58,59], seven potential candidates of GmSnRKs, GmMPKs, GmCDPKs, and GmCKs (Table 1) were tested for interaction with Class II GmACBPs by Y2H screening. The cDNA sequences encoding the soluble domain of GmACBP3.1 and GmACBP4.1 were cloned into the bait vector (pGBKT7, Clontech, Mountain View, CA, USA). The cDNAs of GmACBP3.1 and GmACBP4.1 were amplified through PCR, using specific primer pairs (Appendix A) ML3564/ML3565 and ML3566/ML3567, respectively. Next, the products were inserted into the pGEM-T Easy Vector (Promega, Madison, WI, USA) to generate two plasmids, pAT1094 and pAT1095. Subsequently, 0.9-kb *Eco*RI-*Bam*HI fragments from these plasmids were digested, purified, and inserted into similar sites on plasmid pGBKT7 (Clontech, Mountain View, CA, USA) to create plasmids pAT1103 and pAT1104, respectively.

The cDNA sequences of the protein kinase candidates were inserted into the prey vector (pGADT7, Clontech, Mountain View, CA, USA). Full-length GmCK2β, GmCDPK4, GmMPK2, GmMPK13, GmSRK2I, and two GmSAPK2 cDNAs were amplified with primer pairs (Appendix A) ML3568/ML3569, ML3570/ML3571, ML3572/ML3573, ML3574/ML3575, ML3580/ML3581, ML3576/ML3577, and ML3578/ML3579, respectively, and cloned into the plasmid pGEM-T Easy vector (Promega, Madison, WI, USA) to produce plasmids pAT1096, pAT1097, pAT1098, pAT1099, pAT1102, pAT1100 and pAT1101, respectively. A 0.8-kb *Xma*I-*Xho*I fragment of GmCK2β, a 1.4-kb *Eco*RI-*Xho*I fragment of GmCDPK4, a 1.1-kb *Eco*RI-*Xho*I fragment of GmMPK2, a 1.1-kb *Eco*RI-*Xho*I fragment of GmMPK3, a 1.1-kb *Eco*RI-*Xho*I fragment of GmSRK2I, a 1-kb *Eco*RI-*Xho*I fragment of GmSAPK2-1.1 and a 1.1-kb *Bam*HI-*Xho*I fragment of a second GmSAPK2-2 were digested from the respective derivatives of plasmid pGEM-T Easy vector (Promega, Madison, WI, USA), purified, and inserted into similar sites on pGADT7 (Clontech, Mountain View, CA, USA) to form plasmids pAT1105, pAT1106, pAT1107, pAT1108, pAT1111, pAT1109, and pAT1110, respectively. DNA sequencing was conducted to verify the plasmids. Appendix A contains the specific PCR-primer sequences used for cloning the kinases and Class II GmACBPs cDNAs.

### 4.2. Yeast Two-Hybrid Assay

The Matchmaker Gold Y2H System (Clontech, Mountain View, CA, USA) was used following the manufacturer’s instructions, and the bait and prey constructs were introduced into Y2H Gold Cells (Clontech, Mountain View, CA, USA) using the lithium acetate method [60]. Co-transformed yeast cells were selected on DDO (synthetic dropout medium/-Leu/-Trp) plates at 30 °C for three days and confirmed using colony PCR. Positive colonies were resuspended in sterile 0.9% (*w*/*v*) sodium chloride. For screening protein interactions, serial dilutions (OD_600_ = 0.1, OD_600_ = 0.01 and OD_600_ = 0.001) of the cell suspensions were spotted on DDO and TDO/X/A (synthetic dropout medium/−His/−Leu/−Trp, containing 40 µg mL^−1^ 5-bromo-4-chloro-3-indolyl-α-D-galactopyranoside (X-α-Gal) and 125 ng mL^−1^ aureobasidin A) plates, which were incubated at a temperature of 30 °C for three days before photography [46].

A further Y2H assay was performed for protein kinases, which showed interactions with the full-length soluble domain of Class II GmACBPs to identify domains essential for protein–protein interaction. Following this, two constructs containing either the ANK or ACB domain were made. The first construct contained GmACBP3.1/GmACBP4.1 with only ANK (using primer pair ML3609/ML3610 (Appendix A), as the ANK domain of Class II GmACBPs is entirely identical; only a single construct was produced for the ANK domain). The two other constructs with only the ACB domain contained GmACBP3.1 or GmACBP4.1 using primer pairs ML3564/ML3608 and ML3566/ML3608 (Appendix A), respectively. The X-α-Gal assay was subsequently performed.

### 4.3. Generation of Bimolecular Fluorescence Complementation Screen (BiFC) Constructs

For BiFC analysis, the ORFs coding for the GmMPK2 and GmSAPK2-1 were amplified using primer pairs ML3611/ML3612 and ML3613/ML3614 (Appendix A) and inserted into the *Xba*I-*Bam*HI and *Bam*HI-*Xho*I sites of the pSPYCE-35S vector [61] to produce plasmids pAT1119 and pAT1120, respectively. The previously constructed pSPYNE-35S vector [61], containing the coding region of the native form of GmACBP4.1 (plasmid pAT995; [14]), was used in this study. For the positive and negative controls, the GmLOX1 (*Glycine max* LIPOXYGENASE1) construct (pAT1006, in pSPYCE-35S) and a splice variant of GmACBP3.3 (pAT994, in pSPYNE-35S) previously reported [14] was used. Lung et al. (2022) had confirmed the absence of interaction (no BiFC signal) between the vegetative LOX homolog (VLXB) and some splice variants of Class II GmACBPs. The splice variants produced proteins with the ANK domain truncated. Therefore, one splice variant of Class II GmACBPs (GmACBP3.3) was selected to identify if the ANK domain is essential for protein–protein interaction.

### 4.4. Confocal Laser Scanning Microscopy

For transient expression, the binary plasmids were introduced into *Agrobacterium tumefaciens* strain GV3101 using the freeze–thaw method [62]. Following this step, *A. tumefaciens* containing fusion constructs of nYFP (pAT995, pAT996) and cYFP (pAT1006, pAT1119, pAT1120), along with a mCherry-HDEL marker, were co-infiltrated into 4-week-old *N. benthamiana* leaves. The examination of five combinations was carried out including GmMPK2:cYFP/GmACBP4.1:nYFP and GmSAPK2:cYFP/GmACBP4.1:nYFP (native form of GmACBP4.1), GmMPK2:cYFP/GmACBP3.3:nYFP, GmSAPK2:cYFP/GmACBP3.3:nYFP (splice variant of GmACBP4.1, as negative control) and GmLOX1:cYFP/GmACBP4.1:nYFP (positive control). After two days of agro-infiltration, the Carl Zeiss LSM 980 confocal laser scanning microscope was employed to examine the protein–protein interaction, with consistent acquisition settings maintained for all BiFC analyses. The excitation and emission wavelengths for YFP and mCherry were 514 nm/491–579 nm and 561 nm/570–695 nm, respectively.

### 4.5. Western Blot Assay

*N. benthamiana* leaves were ground into fine powder using liquid nitrogen, and then mixed with protein extraction buffer at a ratio of 1:4 (*w*/*v*). The protein extraction buffer consisted of 50 mM Tris–HCl (pH 7.5), 0.15 M NaCl, 2 mM EDTA, 10% (*v*/*v*) glycerol, 0.2% (*v*/*v*) β-mercaptoethanol, 1% (*v*/*v*) Triton X-100, 2% (*w*/*v*) polyvinylpolypyrrolidone, 1% (*v*/*v*) protease inhibitor cocktail (Sigma-Aldrich, St. Louis, MI, USA) and 1 mM phenylmethylsulfonyl fluoride. Following that, the proteins were clarified via centrifugation at 20,000× *g* for 30 min [14]. The extraction of proteins from yeast cells was performed following the method described by Zhang et al. (2011, [63]). The *N. benthamiana* and yeast proteins were subjected to separation using a 10% SDS-PAGE gel, followed by their transfer onto polyvinylidene difluoride membranes (Pall Life Sciences, New York, NY, USA). Next, the *N. benthamiana* proteins were incubated with rabbit polyclonal antibodies anti-nYFP (1:2000; Agrisera, Seattle, WA, USA) or anti-cYFP (1:5000; Agrisera), while the yeast proteins were incubated with rabbit polyclonal antibody anti-HA (1:5000; Immunoway, Plano, TX, USA), or mouse polyclonal antibody anti-Myc (1:5000; Abclonal, Woburn, MA, USA) overnight at 4 degrees. Finally, horseradish peroxidase-conjugated secondary antibodies (1:10,000; Sigma-Aldrich) were applied at room temperature for 2 h. The signals were observed using an Alliance Q9 Advanced System (UVITEC, Cambridge, UK) and the Pierce ECL Western Blotting Substrate (Thermo Scientific, Waltham, MA, USA).

### 4.6. Recombinant Protein Expression and Purification for In Vitro Kinase Assay

For (His)_6_-tagged proteins, the cDNA sequences encoding the soluble domain of GmACBP4.1, GmMPK2, and GmSAPK2 were amplified using primer pairs ML3150/ML3129, ML3615/ML3616, and ML3617/ML3618 (Appendix A) and cloned into the protein expression vector pRSETA (Invitrogen, Waltham, MA, USA) to produce plasmids pAT963, pAT1121 and pAT1122, respectively. Recombinant proteins were expressed in *Escherichia coli* following Miao et al. (2019; [64]) and purified on immobilized metal affinity (HisTrap HP) and anion-exchange (HiTrap Q HP) chromatography columns (GE Healthcare, Hino, Tokyo) according to Guo et al. (2017; [65]), using an ÄTKA Start FPLC system (Cytiva, Shrewsbury, MA, USA). As the two members of Class II GmACBPs exhibit high homology, only GmACBP4.1 was purified for *in vitro* kinase assay.

### 4.7. In Vitro Kinase Assay and Phos-Tag SDS-PAGE

Class II GmACBPs were further verified to be the substrates of the selected protein kinases, as revealed from Y2H screening and BiFC. As described previously, the purified enzymes and GmACBP4.1 were used for *in vitro* kinase assays [66]. Phos-tag SDS-PAGE (Wako, Osaka, Japan) was run according to the manufacturer’s instructions to detect protein phosphorylation by the mobility shift of protein bands after Coomassie Blue staining. Conventional SDS-PAGE was also performed as a control.

### 4.8. Prediction of Kinase-Specific Phosphorylation Sites in Class II GmACBPs and Docking-Based Binding Models of Class II GmACBPs and Two Kinases

Three highly advanced databases, including the EPSD (the Eukaryotic Phosphorylation Sites Database, http://epsd.biocuckoo.cn/, version 1.0 [67]), GPS (Group-based Phosphorylation, http://gps.biocuckoo.cn/, version 6.0 [68]) and NetPhos (Generic Phosphorylation Sites in Eukaryotic Proteins, http://www.cbs.dtu.dk/, version 3.1 [69]) were used to predict the phosphorylation sites of Class II GmACBPs at serine, threonine and tyrosine residues, as well as the potential kinase families that could catalyze these predicted phosphosites (Appendix A). As the amino acid sequences of GmACBP3.1 and GmACBP4.1 are highly similar, only the GmACBP3.1 protein was used to predict phosphorylation sites. The predictions from the three databases were combined and shown in Figure 5 and Figure 6.

The complexes of GmACBP3.1-GmMPK2 and GmACBP3.1-GmSAPK2 were modeled with the HDOCK server (protein–protein and protein–DNA/RNA docking based on a hybrid algorithm of template-based modeling and ab initio free docking, http://hdock.phys.hust.edu.cn/, [70]). To do this, the tertiary structure models of GmACBP3.1, GmMPK2, and GmSAPK2 were predicted by using the Phyre2 online tool (Protein Homology/analog Recognition Engine, http://sbg.bio.ic.ac.uk/, version 2.0 [71]) and pdb files were submitted in the HDOCK server. Finally, the results page provided access to the top 100 anticipated complex structures from which the top 10 models could be viewed.

## 5. Conclusions

Given the significance of plant ACBPs in various plant growth and development processes, we conducted this study to gain an insight into their potential role in signaling pathways. This study investigated the interaction between two members of Class II GmACBPs and seven kinases using Y2H, BiFC, and *in vitro* kinase assay. The results showed that both GmACBP3.1 and GmACBP4.1 interact with two soybean kinases, GmMPK2 and GmSAPK2. Additionally, this study tested the importance of the ankyrin domain in facilitating protein–protein interactions. Although previous studies have shown the significance of this domain in the interaction between plant ACBPs and other proteins, their role in interactions with kinases has not been reported. This study represents the first report describing the interaction between Class II GmACBPs and kinases. It suggests that plant ACBPs have the potential to function as phospho-proteins to influence signaling pathways in response to harsh environmental conditions.

The current research on the interactors of Class II GmACBPs forms a starting point for future studies to enhance the comprehension of the function of Class II GmACBPs and their interactors in stress responses. Therefore, additional studies and laboratory experiments are necessary to determine how these kinases impact the functions of Class II GmACBPs during different stress conditions, including salinity and drought. Furthermore, it is essential to identify the level of significance of the interaction between GmACBP3.1/GmACBP4.1 and GmMPK2/GmSAPK2 in the stress response, and to discover the phosphorylation status of Class II GmACBPs under various stress conditions.

## Figures and Tables

**Figure 1 plants-13-01146-f001:**
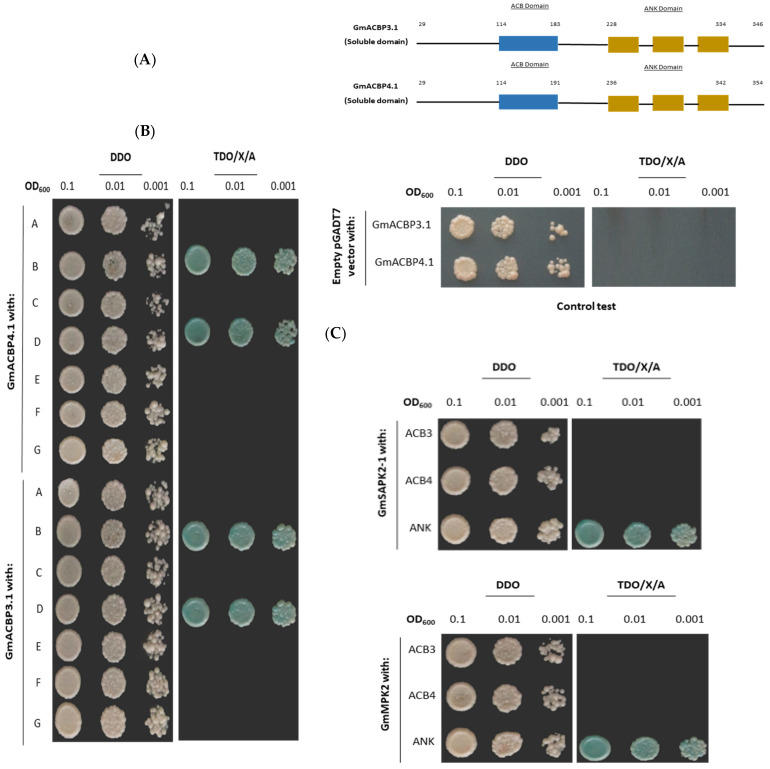
Interactions between the Class II GmACBPs (*Glycine max* ACBPs) and kinases by yeast-two hybrid (Y2H) assay. (**A**) Structure of Class II GmACBPs (GmACBP3.1 and GmACBP4.1). (**B**) The full-length soluble domains of Class II GmACBPs were examined for protein–protein interaction (PPI) with the seven potential candidates of protein kinases. The successful co-transformation and PPI were demonstrated by the presence of white colonies on DDO (synthetic dropout medium/-Leu/-Trp) and blue colonies on TDO/X/A (synthetic dropout medium/−His/−Leu/−Trp containing 40 µg mL^−1^ 5-bromo-4-chloro-3-indolyl-α-D-galactopyranoside (X-α-Gal) and 125 ng mL^−1^ aureobasidin A) plates, respectively. The Y2H experiment demonstrated that Class II GmACBPs interact with both GmMPK2 and GmSAPK2. A, B, C, D, E, F, and G represent CDPK4, MPK2, MPK3, SAPK2-1, SAPK2-2, CK2, and SRK2I kinases, respectively. For the control test, the empty pGADT7 vector was used. (**C**) Further Y2H assay was performed to identify the domains in Class II GmACBPs essential for PPI. The truncated versions of Class II GmACBPs (without the ankyrin-repeat (ANK) and acyl-CoA-binding (ACB) domains) were used for PPI with GmMPK2 and GmSAPK2, and the result showed that both kinases only interacted with the ANK domain, and not with the ACB domain.

**Figure 2 plants-13-01146-f002:**
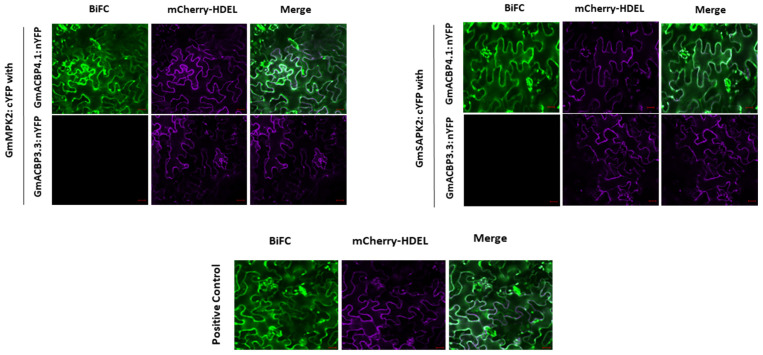
Bimolecular fluorescence complementation (BiFC) shows *in vivo* interaction between kinases and the native form of GmACBP4.1 but not with the GmACBP3.3 splice variant without ANK domain, which demonstrated that only the ANK domain activity of Class II GmACBPs is essential for their interaction with kinases. Confocal laser scanning microscopy was used to examine the signals from BiFC after agroinfiltration *Nicotiana benthamiana* leaf epidermal cells with split-YFP fusion constructs (consisting of nYFP and cYFP) alongside the mCherry-HDEL marker, which served as a transfection control. Positive control (GmLOX1 (*Glycine max* LIPOXYGENASE1) with full-length GmACBP4.1), negative control (splice variant GmACBP3.3 with kinases), bars = 20 μm.

**Figure 3 plants-13-01146-f003:**
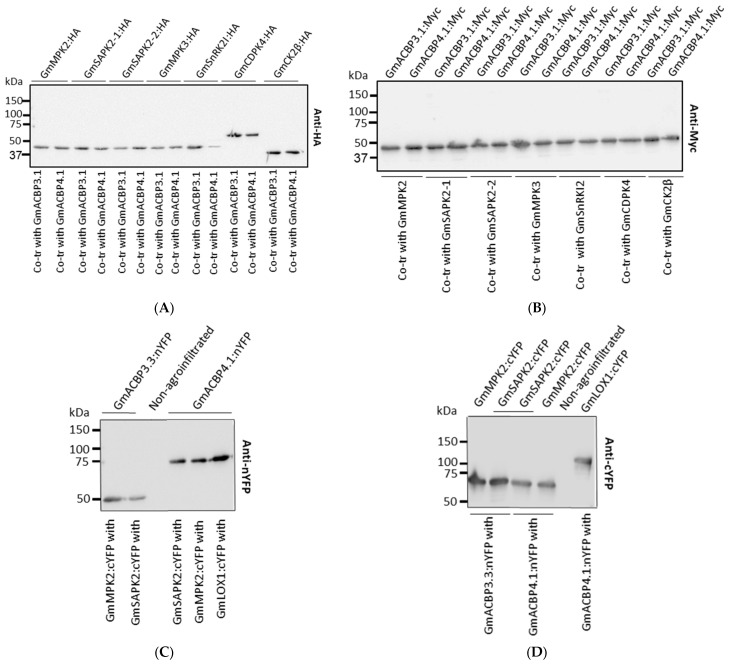
Western blot images of Y2H constructs after co-transformation (**A**,**B**) and BiFC construct combinations (**C**,**D**) to validate the expression of and production from each construct. Total proteins (20 µg/lane) of *N. benthamiana* leaves and yeast cells were resolved on 10% SDS-PAGE and analyzed by Western blot with anti-HA (**A**), anti-Myc (**B**), anti-nYFP (**C**) and anti-cYFP (**D**) antibodies. (**A**) The target bands are 43-kD GmMPK2:HA (apparent: 50 kD), 42-kD GmSAPK2-1:HA (apparent: 50 kD), 42-kD GmSAPK2-2:HA (apparent: 50 kD), 42-kD GmMPK3:HA (apparent: 50 kD), 42-kD GmSnRK2I:HA (apparent: 50 kD), 56-kD GmCDPK4:HA (apparent: 63 kD), 33-kD GmCK2β:HA (apparent: 40 kD). (**B**) The target bands are 36-kD GmACBP3.1:Myc (apparent: 45 kD) and 36-kD GmACBP4.1:Myc (apparent: 45 kD). (**C**) The target bands are 42-kD GmACBP3.3 (splice variant):nYFP (apparent: 50 kD), 59-kD GmACBP4.1 (native):nYFP (apparent: 72 kD). (**D**) The target bands are 52-kD GmMPK2:cYFP (apparent: 62 kD), 51-kD GmSAPK2:cYFP (apparent: 62 kD), and 108-kD GmLOX1:cYFP.

**Figure 4 plants-13-01146-f004:**
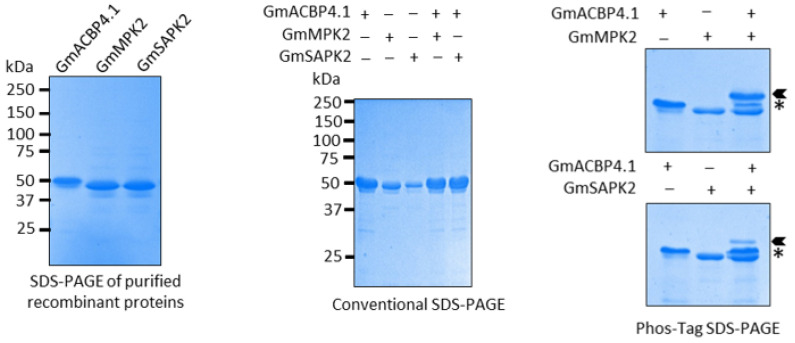
Phosphorylation of Class II GmACBPs analyzed by Phos-tag assay. The purified kinases and GmACBP4.1 were used for *in vitro* kinase assays, and Coomassie Blue-stained Phos-tag gel was used to separate phosphorylated GmACBP4.1 after incubation with GmMPK2 and GmSAPK2. Upon analysis, a shift in size was observed for GmACBP4.1 after treatment with both kinases. Conventional SDS-PAGE was also performed as a control. Arrowheads indicate phosphorylated GmACBP4.1; Asterisks indicate non-phosphorylated GmACBP4.1.

**Figure 5 plants-13-01146-f005:**
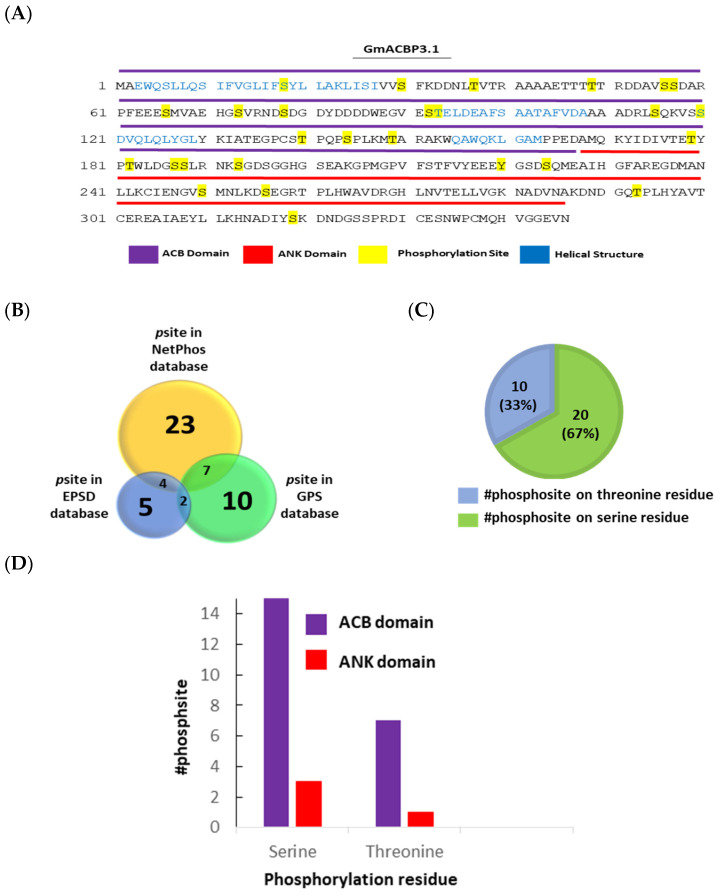
Mapping of the phosphorylation site prediction results of Class II GmACBPs. (**A**) The potential sites for phosphorylation of GmACBP3.1 were predicted using the EPSD (the Eukaryotic Phosphorylation Sites Database, http://epsd.biocuckoo.cn/, version 1.0), GPS (Group-based Phosphorylation, http://gps.biocuckoo.cn/, version 6.0) and NetPhos (Generic Phosphorylation Sites in Eukaryotic Proteins, http://www.cbs.dtu.dk/, version 3.1). Due to the significant similarity in amino acid sequences between GmACBP3.1 and GmACBP4.1, only the GmACBP3.1 protein was utilized for predicting the phosphorylation sites. Yellow indicates the phosphorylation site residues predicted by the combination of database results. Purple, red, and blue represent the ACB domain, ANK domain, and helical structures, respectively. (**B**) A Venn diagram displays the number of predicted phosphorylation sites. The combination of these databases predicted a total of thirty-eight phosphosites. The *p*Site stands for phosphosite(s). (**C**) Prediction results were merged to create a Venn diagram which displays the distribution of phosphorylation on serine and threonine residues. The number of phosphorylation events on serine residues is twice that of threonine residues. (**D**) This chart displays the predicted number of phosphorylation sites on the ACB and ANK domains. The distribution of these phosphosites on Class II GmACBPs showed twenty-four phosphosites on the ACB domain and four on the ANK domain.

**Figure 6 plants-13-01146-f006:**
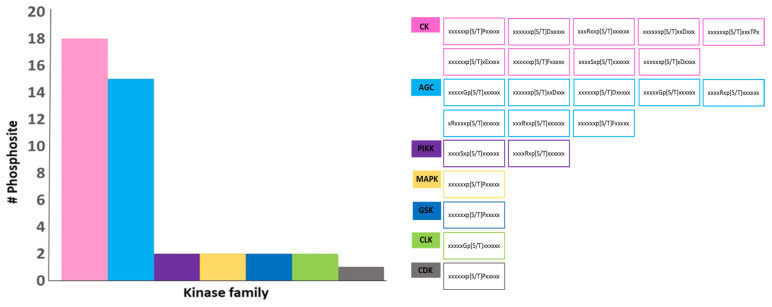
Correlation between the kinase family and the kinase docking site motif of Class II GmACBPs. The databases generated a list of potential kinase families (a total of seven) that could catalyze phosphorylation at the predicted site. The identified kinase families include AGC (cAMP-dependent protein kinase), CK (casein kinase), CDK (cyclin-dependent kinase), MPK (mitogen-activated protein kinase), GSK (glycogen synthase kinase), PIKK (phosphatidylinositol-3 kinase-related kinase) and CLK (Cdc2-like kinase). The (**left panel**) shows the number of phosphorylation site motifs associated with each kinase family, and the (**right panel**) displays the motifs (38) linked with each kinase family. Among the potential kinases identified, the CK and AGC kinase families possess the highest number of motifs, and several families of kinases possess similar docking-site motifs.

**Figure 7 plants-13-01146-f007:**
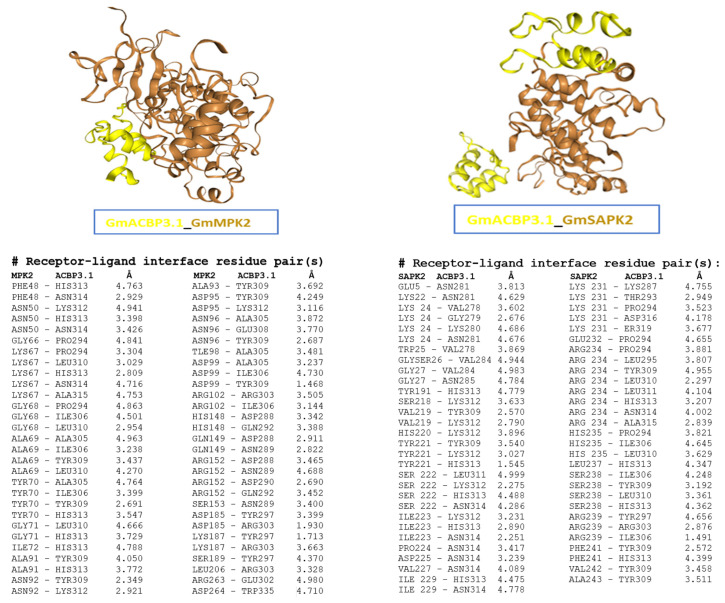
Docking models of the GmACBP3.1-GmMPK2 and GmACBP3.1-GmSAPK2 complexes, generated by the HDOCK server. The models with the lowest docking scores and highest confidence scores are depicted. GmACBP3.1, yellow, and kinases, brown, are in ribbon representation (**upper panel**). The (**lower panel**) shows binding site residues on both the receptor (kinases) and ligand (GmACBP3.1) and distance restraints between the binding site residues of the receptor and ligand. Å: the angstrom is a metric unit of length/distance, and one angstrom equals 10^−10^ m.

## Data Availability

Data are contained within the article and Appendix A.

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
