# Peer review of "Interaction of Soybean (Glycine max (L.) Merr.) Class II ACBPs with MPK2 and SAPK2 Kinases: New Insights into the Regulatory Mechanisms of Plant ACBPs"

_plants, 2024, doi:10.3390/plants13081146_

Round 1
Reviewer 1 Report
Comments and Suggestions for Authors
In the manuscript ‘Soybean Class II Acyl-CoA-Binding Proteins Interact With Mitogen-Activated Protein Kinase MPK2 and Serine/Threonine-Protein Kinase SAPK2’, Moradi and collaborators described the interaction between Class II GmACBPs and kinases with the aim to understand their interactions in plant stress responses.
Overall, the manuscript is well written, it is written in an appropriate way and the English language is appropriate, clear and understandable. The manuscript is well organized. The Materials and Methods section describes clearly the methodologies used which are also appropriate to accomplish the objectives of the study. The data and analyses are presented appropriately. The legends are very detailed. The Discussion section makes an appropriate discussion of the results obtained and the authors compare their results with the findings obtained in other related works. The authors present the conclusions of their work and they also point out further works.
Author Response
Response to Reviewer 1
Comments and Suggestions for Authors:
In the manuscript ‘Soybean Class II Acyl-CoA-Binding Proteins Interact With Mitogen-Activated Protein Kinase MPK2 and Serine/Threonine-Protein Kinase SAPK2’, Moradi and collaborators described the interaction between Class II GmACBPs and kinases with the aim to understand their interactions in plant stress responses.
Overall, the manuscript is well written, it is written in an appropriate way and the English language is appropriate, clear and understandable. The manuscript is well organized. The Materials and Methods section describes clearly the methodologies used which are also appropriate to accomplish the objectives of the study. The data and analyses are presented appropriately. The legends are very detailed. The Discussion section makes an appropriate discussion of the results obtained and the authors compare their results with the findings obtained in other related works. The authors present the conclusions of their work and they also point out further works.
Response: Thank you very much for taking the time to review this manuscript and for your positive and detailed review of the manuscript.

Reviewer 2 Report
Comments and Suggestions for Authors
The paper by Moradi et al., is written well and present novel results for the interaction between Class II GmACBPs and kinases. However some minor changes are required before formal acceptance of the paper.
Generally the study is scientifically sound and new results are achieved.
Add scientific name of the targeted species in the title.
Though the title provide a sharp interpretation of the results, it’s better to highlight the role of this interaction in soybean.
Add a brief hypothesis of the study that why you have studied the interaction of these proteins?
References are not according to the journal format, revise it.
Some sections like 2.2, 2.4, 2.5, 2.6 are missing references.
Combine figure 1 into one figure and label it with Alphabets. Moreover same font style should be applied to each figure.
Results are presented well, however minor English correction is required.
Add a summary paragraph to the discussion section.
Present the conclusion of the study with a brief explanation that what are your objectives and how you have achieved it.
Comments on the Quality of English LanguageMinor English editing is required.
Author Response
Response to Reviewer 2
Comments and Suggestions for Authors:
The paper by Moradi et al., is written well and present novel results for the interaction between Class II GmACBPs and kinases. However some minor changes are required before formal acceptance of the paper. Generally the study is scientifically sound and new results are achieved
Response 1: We sincerely appreciate your valuable time and effort in reviewing this manuscript.
Add scientific name of the targeted species in the title.
Response 2: The scientific name of soybean was added in the title: “Interaction of soybean (Glycine max (L.) Merr.) Class II ACBPs with MPK2 and SAPK2 kinases: new insights into the regulatory mechanisms of plant ACBPs”.
Though the title provide a sharp interpretation of the results, it’s better to highlight the role of this interaction in soybean.
Response 3: Some sentences to highlight the role of this interaction in soybean have been added in the Discussion section: “The evidence suggests that MPK2 and SAPK2 kinases are essential in signaling pathways of various plants, especially in soybean. Earlier studies [35,43,45] have demonstrated their interaction with protein partners of plant ACBPs such as AREB1 and RELATED TO AP2.12, while the current study illustrates their interaction with Class II GmACBPs. This implies that these kinases have the potential to affect the function of Class II GmACBPs. These interactions provide new insights into the regulatory mechanisms of plant ACBPs. It suggests these interactions may affect soybean growth, development, and response to environmental stresses, including biotic and abiotic factors. Furthermore, an understanding of these interactions holds potential implications for soybean breeding, genetic engineering, and crop improvement efforts.” (Lines 425-434).
Add a brief hypothesis of the study that why you have studied the interaction of these proteins?
Response 4: we have incorporated several sentences regarding why we have studied the interaction of these proteins in the Discussion section: “It is known that ACBPs play crucial roles in lipid metabolism, stress responses, and hormone signaling pathways in plants. Given the importance of protein-protein interactions in signaling pathways, we hypothesized that the interaction between Class II GmACBPs and kinases could potentially play a role in modulating these pathways. By investigating the interaction between Class II GmACBPs and specific kinases, we aim to gain insights into the potential involvement of plant ACBPs in sig-naling cascades and the underlying mechanisms that contribute to plant growth and development. An understanding on these interactions could provide valuable infor-mation for future research on manipulating signaling pathways and improving plant traits for various agricultural applications”. (Lines 330-339).
References are not according to the journal format, revise it.
Response 5: The references have been revised according to the journal format.
Some sections like 4.2, 4.4, 4.5, and 4.6 are missing references.
Response 6: References have been added for sections 4.2, 4.4, and 4.5. Additionally, prior to the revision step, references for protein expression and purification in section 4.6 were included.
Combine figure 1 into one figure and label it with Alphabets. Moreover same font style should be applied to each figure.
Response 7: Two photos of Figure 1 have been combined and labeled with Alphabets. Additionally, the font style of all Figures has been checked.
Results are presented well, however minor English correction is required.
Response 8: English correction has been applied to all parts of the manuscript.
Add a summary paragraph to the discussion section.
Response 9: prior to the revision step, some sentences as a summary were included in the Discussion section: “This study used several bait and prey plasmids for Y2H analysis to investigate the interaction between Class II GmACBPs and seven kinases (Figure 1). It utilized BiFC to confirm the identified interacting partners (Figure 2). Class II GmACBPs were found to interact with two soybean kinases, namely GmMPK2 and GmSAPK2. Also, this study showed that these two kinases can phosphorylate Class II GmACBPs in vitro (Figure 4). Experiments in Y2H, BiFC, and docking modeling studies demonstrated that the interactions were facilitated by the ankyrin domain. These interactions provide essential clues for a more comprehensive understanding of the role of plant ACBPs in signaling pathways”. (Lines 340-348).
.
Present the conclusion of the study with a brief explanation that what are your objectives and how you have achieved it.
Response 10: The Conclusion section has been added.
Comments on the Quality of English Language:
Minor English editing is required.
Response 11: English correction has been applied to all parts of the manuscript.

Reviewer 3 Report
Comments and Suggestions for Authors
Dear authors and the editor,
This an interesting research focus on the interaction test of Glycine max ACBPs and seven kinases in soybean, and the result showed that MPK2 and SAPK2 could interact with ACBPs based on Y2H and BiFC experiments. This study may be helpful for explaining why the phosphate fertilizer could improve drought and tolerance tolerance, and as well as providing new evidence of understanding ACBPs regulation mechanism response with envirmental factors. But there are some poor presentation in title which is too long, abbreviation might be used and please show this protein function. Second, some figures such as Fig 2, 5, 6 and 7 lack of high quality in expression and clear. Finally, please provide some highlights in abstract.
Comments on the Quality of English LanguageDear authors and the editor,
This an interesting research focus on the interaction test of Glycine max ACBPs and seven kinases in soybean, and the result showed that MPK2 and SAPK2 could interact with ACBPs based on Y2H and BiFC experiments. This study may be helpful for explaining why the phosphate fertilizer could improve drought and tolerance tolerance, and as well as providing new evidence of understanding ACBPs regulation mechanism response with envirmental factors. But there are some poor presentation in title which is too long, abbreviation might be used and please show this protein function. Second, some figures such as Fig 2, 5, 6 and 7 lack of high quality in expression and clear. Finally, please provide some highlights in abstract.
Author Response
Response to Reviewer 3
Comments and Suggestions for Authors:
This an interesting research focus on the interaction test of Glycine max ACBPs and seven kinases in soybean, and the result showed that MPK2 and SAPK2 could interact with ACBPs based on Y2H and BiFC experiments. This study may be helpful for explaining why the phosphate fertilizer could improve drought and tolerance tolerance, and as well as providing new evidence of understanding ACBPs regulation mechanism response with envirmental factors.
Response 1: We deeply appreciate your valuable time and effort in reviewing this manuscript. Your feedback has been instrumental in improving the quality of our work.
But there are some poor presentation in title which is too long, abbreviation might be used and please show this protein function.
Response 2: The manuscript title has been revised to accurately reflect the function of plant ACBPs, and abbreviations have been used to shorten it: “Interaction of soybean (Glycine max (L.) Merr.) Class II ACBPs with MPK2 and SAPK2 kinases: new insights into the regulatory mechanisms of plant ACBPs”
Second, some figures such as Fig 2, 5, 6 and 7 lack of high quality in expression and clear.
Response 3: The quality of Figures has been improved.
Finally, please provide some highlights in abstract.
Response 4: Four highlights have been added in the Abstract section.

Reviewer 4 Report
Comments and Suggestions for Authors
Overall, the manuscript was well written with clear objectives and conclusions through which GmACBP3.1 and 4.1 have been identified as interacting partners and potential substrates of GmMPK2 and GmSAPK2 kinases. Their interactions via ankyrin domain was convincingly demonstrated in their experiments including Y2H, BiFC, and docking modeling study, so this piece of information alone warrants publication of this manuscript in the journal Plants.
However, this reviewer is somewhat skeptical in their in vitro kinase assay data despite the data itself showed a clear kinase activities of GmMPK2 and GmSAPK2 on the GmACBP4.1 because the proteins used in the assays were all expressed and purified from E.coli, thus it is highly unlikely that these kinases would have remain in active forms. The Phos-Tag SDS-PAGE data in Figure 4 should have included an irrelevant substrate protein as a negative control for the kinase activities to address this issue.
Author Response
Response to Reviewer 4
Comments and Suggestions for Authors:
Overall, the manuscript was well written with clear objectives and conclusions through which GmACBP3.1 and 4.1 have been identified as interacting partners and potential substrates of GmMPK2 and GmSAPK2 kinases. Their interactions via ankyrin domain was convincingly demonstrated in their experiments including Y2H, BiFC, and docking modeling study, so this piece of information alone warrants publication of this manuscript in the journal Plants.
Response 1: We greatly appreciate your effort in reviewing this manuscript.
However, this reviewer is somewhat skeptical in their in vitro kinase assay data despite the data itself showed a clear kinase activities of GmMPK2 and GmSAPK2 on the GmACBP4.1 because the proteins used in the assays were all expressed and purified from E.coli, thus it is highly unlikely that these kinases would have remain in active forms.
Response 2: We hold a different opinion from the reviewer's skepticism concerning the in vitro kinase assay data. The data undeniably illustrates the kinase activities of GmMPK2 and GmSAPK2 on GmACBP4.1. Numerous publications have successfully employed recombinant kinases expressed and purified in E. coli, similar to our work, and these have consistently yielded reliable results. For your review, we have included two papers as examples, which expressed similar kinases in E. coli as our study and followed by in vitro kinase assays (Note that MPK6 and SnRK2.8 are Arabidopsis homologs of soybean MPK2 and SAPK2, respectively). Additionally, it is crucial to utilize freshly-purified kinases for in vitro kinase assays, avoiding long-term storage and freeze-thaw cycles to maintain their activity. We have taken care to do so.
Bahk, S., Ahsan, N., An, J., Kim, S.H., Ramadany, Z., Hong, J.C., Thelen, J.J. and Chung, W.S., 2024. Identification of mitogen-activated protein kinases substrates in Arabidopsis using kinase client assay. Plant Signaling & Behavior, 19(1), p.2326238.
Shin, R.; Alvarez, S.; Burch, A.Y.; Jez, J.M.; Schachtman, D.P. Phosphoproteomic identification of targets of the Arabidopsis sucrose nonfermenting-like kinase SnRK2.8 reveals a connection to metabolic processes. Proc Natl Acad Sci USA. 2007, 104, 6460-6465.
The Phos-Tag SDS-PAGE data in Figure 4 should have included an irrelevant substrate protein as a negative control for the kinase activities to address this issue.
Response 3: We appreciate the reviewer's suggestion to include an irrelevant substrate protein as a negative control for the kinase activities in Figure 4. However, we believe that the two in vitro kinase reactions we used as controls, one containing only GmACBP4.1 and the other containing only kinases, are sufficient to support our findings. We observed no shift in size for GmACBP4.1 in the control reactions, unlike in the in vitro kinase reaction that contained both GmACBP4.1 and kinases. Therefore, we feel that these two controls adequately validate the results of the in vitro kinase assay and Phos-Tag SDS-PAGE.

Round 2
Reviewer 2 Report
Comments and Suggestions for Authors
The authors addressed all the concerns.
Reviewer 3 Report
Comments and Suggestions for Authors
Dear authors and the editor,
Most problems have been modified and improved, so I suggest publicate this MS in present form.
Comments on the Quality of English LanguageDear authors and the editor,
Most problems have been modified and improved, so I suggest publicate this MS in present form.